# Searching Parameterized AP Loss
# for Object Detection

**Chenxin Tao**[1][†][*], **Zizhang Li**[2][†][*], **Xizhou Zhu**[3][*], **Gao Huang**[1], **Yong Liu**[2], **Jifeng Dai**[3,4][‡]
[1]Tsinghua University, [2]Zhejiang University, [3]SenseTime Research,
[4]Shanghai Jiao Tong University
tcx20@mails.tsinghua.edu.cn, zzli@zju.edu.cn,
{zhuwalter, daijifeng}@sensetime.com
gaohuang@tsinghua.edu.cn, yongliu@iipc.zju.edu.cn

## Abstract

Loss functions play an important role in training deep-network-based object detectors. The most widely used evaluation metric for object detection is Average Precision (AP), which captures the performance of localization and classification sub-tasks simultaneously. However, due to the non-differentiable nature of the AP metric, traditional object detectors adopt separate differentiable losses for the two sub-tasks. Such a mis-alignment issue may well lead to performance degradation. To address this, existing works seek to design surrogate losses for the AP metric manually, which requires expertise and may still be sub-optimal. In this paper, we propose Parameterized AP Loss, where parameterized functions are introduced to substitute the non-differentiable components in the AP calculation. Different AP approximations are thus represented by a family of parameterized functions in a unified formula. Automatic parameter search algorithm is then employed to search for the optimal parameters. Extensive experiments on the COCO benchmark with three different object detectors (*i.e.,* RetinaNet, Faster R-CNN, and Deformable DETR) demonstrate that the proposed Parameterized AP Loss consistently outperforms existing handcrafted losses. Code shall be released.

## 1 Introduction

The past decade has witnessed the significant success of deep neural networks in object detection, in which loss functions play an indispensable role in training networks. To evaluate the object detection methods, the Average Precision (AP) metric is usually used, which captures the performance of localization and classification simultaneously. However, as in most object detectors [26], the training of localization and classification sub-tasks are driven by two separate losses (see Figure 1). For example, the L1/smooth-L1 [11] or GIoU [38] losses are usually employed for localization, while the cross-entropy or Focal [25] losses are usually used for classification. Such a mis-alignment between network training and evaluation may well lead to performance degradation.

To mitigate this mis-alignment issue, a straight-forward solution is to approximate the AP metric in network training. Because the AP metric is non-differentiable, many works [3, 5, 13, 40, 35] have explored hand-crafted losses based on the mathematical formula of the AP metric. [3] replaces the non-differentiable parts in the AP metric with hand-crafted differentiable approximations. The loss gradient is obtained by taking derivation with respect to the hand-crafted function. However, as both [5] and our experiments show, such a hand-crafted AP approximated loss produces lower

---

[*]Equal contribution. [†]This work is done when Chenxin Tao and Zizhang Li are interns at SenseTime Research. [‡]Corresponding author.

35th Conference on Neural Information Processing Systems (NeurIPS 2021).

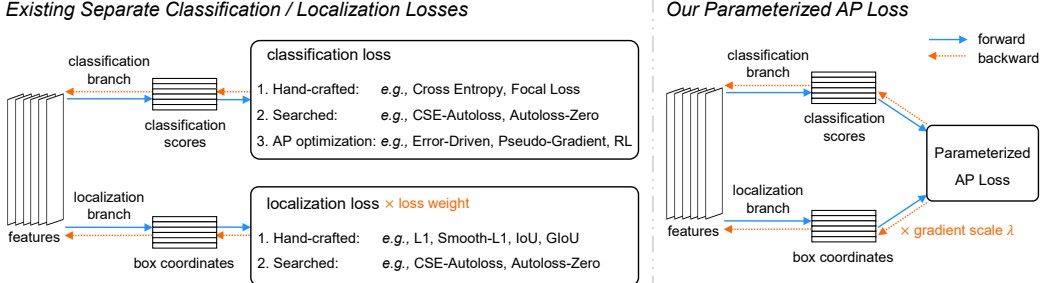

**Figure 1:** Comparison between existing approaches and our Parameterized AP Loss.

performance than the commonly used cross-entropy and L1/smooth-L1 losses. Another line of works try to estimate the gradient for the AP metric directly [5, 13, 40, 35]. These works try to manually design the loss gradient. Therein, the AP approximated gradient only drives the training of the classification branch, while the training of the localization branch is still supervised by traditional regression losses. In practice, these methods still do not address the mis-alignment issue well.

The common issue of existing approaches is they do not optimize over the numerous approximations of the non-differentiable discrete AP metric. The AP metric itself is a piecewise constant function, whose differentiable approximations have infinite possibilities. The hand-crafted approximations / gradients may well be sub-optimal for driving network training. Instead of manually determining the surrogate loss form, we propose to approximate the non-differentiable parts with a family of continuous parameterized functions, which helps to represent the numerous AP approximations in a unified formula. Then, an efficient parameter search procedure is employed to find out the desired loss function, so as to optimize the trained object detector's performance on the evaluation set with the AP metric. Because the parameterized AP approximations constitute a compact search space, the search process would be very effective.

To this end, we propose the Parameterized AP Loss, which is built on top of the AP metric to mitigate the mis-alignment between network training and evaluation. It utilizes parameterization to construct the search space so that the optimal parameters can be searched automatically. Specifically, we first explicitly reformulate the AP metric for object detection as a function of classification scores and box coordinates. Then we replace the non-differentiable components in the reformulated function with parameterized functions. Finally, to obtain the optimal loss function for network training, we search the parameters through a reinforcement-learning-based search process, which aims to maximize the AP score on the evaluation set.

We evaluate the searched Parameterized AP Loss on various object detectors, including RetinaNet [25], Faster R-CNN [37] and Deformable DETR [42]. Extensive experiments on the COCO benchmark [23] demonstrate that the proposed Parameterized AP Loss consistently outperforms existing delicately designed losses.

The main contributions of our work can be summarized as:

- By reformulating the AP metric and introducing differentiable parameterized substitutions, Parameterized AP Loss can represent numerous AP approximations in a unified formula, which captures classification and localization sub-tasks simultaneously in a single loss function.
- Instead of hand-crafting AP losses or gradient approximations, our approach automatically searches for the optimal parameter, optimizing for the trained object detector performance.
- Extensive experiments on different object detectors demonstrate that the searched Parameterized AP Loss consistently outperforms the existing losses.

## 2 Related work

**Hand-crafted Loss Functions for Object Detection.** Designing loss functions has been an active direction in the field of object detection for long. Cross-entropy and smooth-L1 losses are widely used for the classification and localization sub-tasks, respectively [26]. For the classification sub-task, to mitigate the imbalance problem in one-stage detectors, Focal Loss [25] and GHM [17] propose to adjust the weight of loss or gradient for each predicted box. DR Loss [34] proposes to convert the classification problem into a ranking problem. For the localization sub-task, a series of

works [38, 41] introduce IoU variants for better localization. GFL [22] and GFLv2 [21] point out that the classification and localization branches are separately trained but compositely used in inference, so they merge the localization representation into the classification branch for a joint optimization. But such combination is still not consistent with the AP metric.

Recent works [5, 30] have noticed the mis-alignment issue between network training and evaluation, and try to address it with specifically designed surrogate loss functions. AP Loss [5] replaces the classification task with a ranking task derived from the AP metric via a hand-crafted error-driven gradient. aLRP Loss [30] extends the framework of AP Loss, and unifies classification and localization losses under the LRP metric [29].

These loss functions are all hand-crafted, and may well be sub-optimal in guiding network training. In contrast, our proposed Parameterized AP Loss is automatically searched, which can better align the network training and evaluation.

**Direct Optimization for the AP Metric.** Pioneer works [13, 40, 35] have also tried to use the AP metric as training objective. Due to the non-differentiable nature of the AP metric, these works adopt different methods to estimate the back-propagated gradient. [13] uses finite difference and linear envelope to derive the pseudo-gradients. [40] applies the loss-augmented inference for the gradient estimation. [35] resorts to reinforcement learning for policy gradient.

While these works take a step towards direct optimization for the AP metric, they all focus on the gradient for the classification task, ignoring the localization task, which may well be sub-optimal for the optimization of AP metric. By contrast, our proposed Parameterized AP Loss deals with these two tasks simultaneously in a single loss function, and thus better align training and evaluation processes.

Another line of works try to approximate the AP metric via interpolation or neural network[27, 32, 8]. [27] refactors the computation process and interpolates the loss value with differentiable functions. [32] learns an embedding for prediction and target, so the Euclidean distance between them approximates the metric value. [8] trains a network for sorting operation, so as to construct a differentiable surrogate. Although these methods use differentiable functions to approximate AP metric, they ignore the training process. Our proposed Parameterized AP Loss, however, is searched to directly optimize the AP metric during evaluation, and thus consider the training behavior of loss function. This leads to a more effective loss function for network training.

**Searching Loss Functions for Object Detection.** Recent works [28, 19] have also tried to search suitable loss functions for object detection. In these methods, loss functions are formulated as computational graphs composed of basic mathematical operators. Since the combinations of operators are randomly chosen, the search space is very sparse with a large number of unpromising loss functions. Thus, they have to design specific techniques to accelerate the search process. [28] also relies on hand-crafted initialization. Nevertheless, their searched loss functions only gain marginal improvement on detection tasks. On the other hand, these methods search for separate losses for the localization and classification sub-tasks, thus the mis-alignment between network training and evaluation still exists.

Our proposed Parameterized AP Loss constitutes a compact search space, where different loss functions are represented by different parameters. Since the parameterized search space is continuous, the search process would be more effective than the discrete combinatorial optimization in [28, 19]. Moreover, Parameterized AP Loss drives the localization and classification training with a single unified loss, which better mitigates the mis-alignment issue.

**Hyper-Parameter Optimization.** Previously, grid search or random search [1] are commonly used for hyper-parameter tuning. As the search space becomes more complex, many efficient hyper-parameter optimization methods have been proposed. Bayesian optimization [2, 14] aims to build a prediction model from history data and use it to select the most promising hyper-parameters for evaluation. Bandit-based methods [15, 20, 10] view it as a resource allocation problem and allocate most of the computational resource to the most promising hyper-parameters. Evolutionary algorithms [36, 33] evolve the optimal hyper-parameters from a population of models. Reinforcement learning techniques [43] have also been used to explore the search space via sampling candidates and adjusting the sampling policy.

The introduced parameters in our proposed Parameterized AP Loss are indeed hyper-parameters, which fit the hyper-parameter optimization methods. In this work, we adopt reinforcement learning

to search for the optimal parameters because of its simplicity and efficiency. Other efficient hyper-parameter optimization algorithms can also be employed.

## 3 Method

In this section, we present how Parameterized AP Loss is constructed to better align the training target with the evaluation metric. The overview of our method is illustrated in Figure 2.

### 3.1 Revisiting AP Metric

AP metric is the most widely used evaluation metric for object detection. Given an image, we assume an object detector outputs $N$ detected bounding boxes for each category as $\mathcal{B} = \{(b_i, s_i)\}_{i=1}^N$. Here, $b_i \in \mathbb{R}^4$ denotes the box coordinates of the $i$-th prediction, and $s_i \in \mathbb{R}$ is its corresponding classification score. In the AP metric, these predictions will be matched with a set of ground-truth bounding boxes $\mathcal{G}$. Each prediction will be assigned with zero or one ground-truth bounding box. Those predictions assigned to ground-truth bounding boxes constitute the positive set $\mathcal{P}$, while other predictions form the negative set $\mathcal{N}$. Then, the AP metric score is defined as the area under the precision-recall curve. Following the above notations, it can be written as

$$\text{AP} = \frac{1}{|\mathcal{P}|} \sum_{i \in \mathcal{P}} p(i), \tag{1}$$

where $p(i)$ denotes the precision of the predictions, whose classification scores are higher than that of the $i$-th prediction. Following [5, 3], Eq. (1) can be explicitly calculated as the function of classification scores

$$\text{AP}(\{s_i\}_{i=1}^N) = \frac{1}{|\mathcal{P}|} \sum_{i \in \mathcal{P}} 1 - \frac{\text{rank}^-(s_i)}{\text{rank}(s_i)} = \frac{1}{|\mathcal{P}|} \sum_{i \in \mathcal{P}} 1 - \frac{\sum_{j \in \mathcal{N}} H(s_j - s_i)}{1 + \sum_{j \in \mathcal{B}, j \neq i} H(s_j - s_i)}, \tag{2}$$

where $\text{rank}^-(s_i)$ and $\text{rank}(s_i)$ denote the ranks of the classification score $s_i$ among the negative prediction set $\mathcal{N}$ and the whole prediction set $\mathcal{B}$, respectively. $H(\cdot)$ is the Heaviside step function,

$$H(x) = \begin{cases} 1, & \text{if } x > 0, \\ 0, & \text{otherwise.} \end{cases} \tag{3}$$

Eq. (2) explicitly includes the classification scores into the AP calculation, which helps previous works [5, 3] to explore surrogate losses for the classification task. However, the localization task is also very important for object detection, which is just implicitly contained in Eq. (2).

### 3.2 Parameterized AP Loss

Instead of hand-crafting AP approximations for training networks, we propose to represent the numerous potential AP approximations in a unified parameterized formula, which is denoted as the Parameterized AP Loss. Specifically, we first explicitly reformulate the AP metric as a function of classification scores and box coordinates $\{(b_i, s_i)\}_{i=1}^N$ output by the detector. Then, the non-differentiable components in the reformulated function are substituted with parameterized functions, so as to extend the AP metric to differentiable parameterized approximations.

**Reformulating AP metric as Function of Classification Scores and Box Coordinates.** Since Eq. (2) is only related to classification scores explicitly, it needs to be further extended as the function of box coordinates. In Eq. (2), the only part related to the localization task for object detection is the summation range (*i.e.,* the positive set $\mathcal{P}$ and the negative set $\mathcal{N}$), which motivates us to replace the summation range by explicitly including the localization results.

To this end, we define the localization score for the $i$-th prediction as

$$l(b_i) = \begin{cases} \text{IoU}(b_i, b_{i^*}), & i \in \mathcal{P}, \\ 0, & i \in \mathcal{N}, \end{cases} \tag{4}$$

where $i^*$ denotes the assigned ground-truth bounding box to the $i$-th prediction, and $\text{IoU}(b_i, b_{i^*})$ calculates the area overlap ratio between these two boxes. Here, box IoU is used to measure the

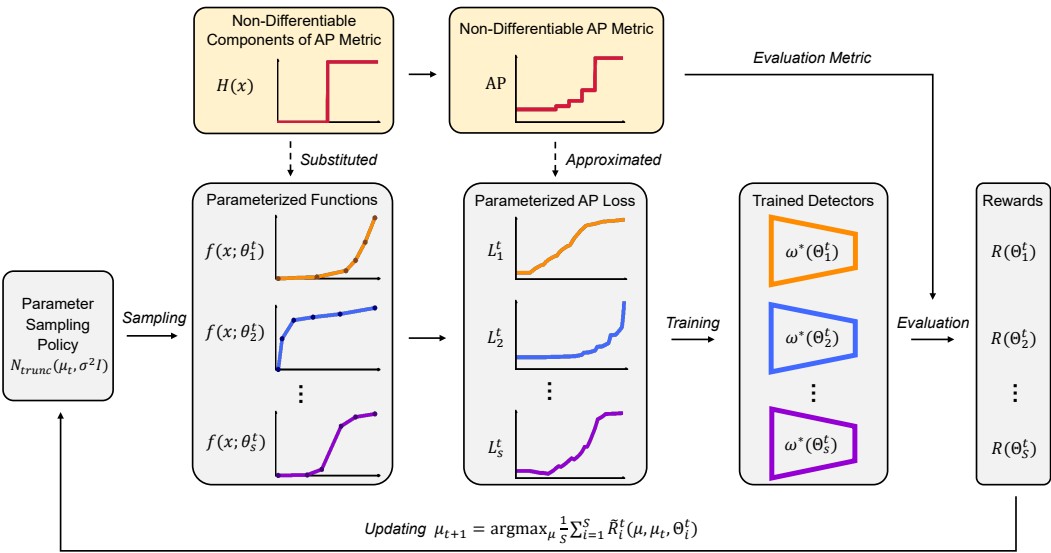

**Figure 2:** Overview of searching Parameterized AP Loss for object detection.

localization results, which follows the AP metric. Other measurements, such as GIoU [38] and L1 are also applicable. Results in Section 4.3 demonstrate that GIoU actually delivers the best results.

By further replacing the summation range in Eq. (2) using Eq. (4), the AP metric can be explicitly formulated as the function of classification scores and box coordinates:

$$\text{AP}(\{(b_i, s_i)\}_{i=1}^N) = \frac{1}{|\mathcal{P}|} \sum_{i \in \mathcal{B}} H\big(l(b_i)\big) - \frac{\sum_{j \in \mathcal{B}, j \neq i} H(s_j - s_i)\big(1 - H\big(l(b_j)\big)\big)}{1 + \sum_{j \in \mathcal{B}, j \neq i} H(s_j - s_i)} H\big(l(b_i)\big). \quad (5)$$

Here, $H\big(l(b_i)\big)$ is the Heaviside step function with the localization score as input, which indicates whether the $i$-th prediction belongs to the positive set. We multiply each summation term with the appropriate indicator function to replace the summation range.

**Extending AP metric to Differentiable Parameterized Approximations.** Designing approximated loss functions for the AP metric may well mitigate the mis-alignment between network training and evaluation. We need to replace the non-differentiable Heaviside step function $H(\cdot)$ in Eq. (5) with its differentiable substitutions. Previous works [5, 30] have tried to address this problem with a specially designed error-driven update technique. Nevertheless, these hand-crafted methods are all based on heuristics and expertise, which may not be optimal to guide the training process.

In contrast to hand-crafting differentiable surrogates for the AP metric, we substitute the non-differentiable components with parameterized functions, which helps to represent different AP approximations in a unified formula. To achieve this, we replace the non-differentiable Heaviside step function $H(\cdot)$ in Eq. (5) with parameterized functions $f(\cdot; \theta)$ parameterized by $\theta$. Then, the proposed Parameterized AP Loss can be obtained as

$$L = -\frac{1}{|\mathcal{P}|} \sum_{i \in \mathcal{B}} f(l(b_i); \theta_1) - \frac{\sum_{j \in \mathcal{B}, j \neq i} f(s_j - s_i; \theta_2)(1 - f(l(b_j); \theta_3))}{1 + \sum_{j \in \mathcal{B}, j \neq i} f(s_j - s_i; \theta_4)} f(l(b_i); \theta_5). \quad (6)$$

Note that we normalize the input of $f(x; \theta)$ to $x \in [0, 1]$ for a unified domain (see Section 4.1 for implementation details). The output range is also restricted to $f(x; \theta) \in [0, 1]$, which follows the value range of $H(\cdot)$. We adopt different parameters $\{\theta_i\}_{i=1}^5$ in Eq. (6) to substitute different $H(\cdot)$ functions in Eq. (5), which makes the Parameterized AP Loss more flexible. Our experiments in Section 4.3 also demonstrate that such parameterization delivers better performance than that of employing shared parameters for different $H(\cdot)$ functions. We also block the gradient of $f(s_j - s_i; \theta_4)$ on the denominator, and empirically find such modification brings more stable training (see Appendix A.1).

**Piecewise Linear Function.** The parameterized function $f(x; \theta)$ can be of any family of differentiable functions. Here, we adopt the piecewise linear function for simplicity. The piecewise linear function is composed of linear segments over different input ranges, where the first point of one

segment coincides with the last point of the previous segment. Assuming a piecewise linear function $f(x; \theta)$ has $M$ segments, the $k$-th segment is defined as

$$f_k(x; \theta) = \frac{y_{k+1} - y_k}{x_{k+1} - x_k} \cdot (x - x_k) + y_k, \quad x_k \leq x < x_{k+1}, \quad k = 0, \ldots, M-1. \quad (7)$$

The $k$-th segment is controlled by two points $(x_k, y_k)$ and $(x_{k+1}, y_{k+1})$. The coordinates of these control points build up the set of parameters $\theta$.

Following [18], we apply the end-point constraint and monotonicity constraint to regularize the search space. The end-point constraint requires that the end points of the parameterized function should take the same value with the original function, while the monotonicity constraint requires the parameterized function to be monotonically increasing. These constraints can be easily applied via

$$x_0 = 0, \ y_0 = 0, \quad x_M = 1, \ y_M = 1; \quad \text{(end-point constraint)}$$

$$0 < \frac{x_k - x_{k-1}}{x_M - x_{k-1}} < 1, \quad 0 < \frac{y_k - y_{k-1}}{y_M - y_{k-1}} < 1, \quad k = 1, \ldots, M-1. \quad \text{(montonicity constraint)}$$

To apply the above restrictions in optimization, the specific form of the parameters is defined as

$$\theta = \left\{ \left( \frac{x_k - x_{k-1}}{x_M - x_{k-1}}, \frac{y_k - y_{k-1}}{y_M - y_{k-1}} \right), k = 1, \ldots, M-1 \right\}. \quad (8)$$

Such parameterization also makes each parameter independent, which simplifies the search.

**Gradient Scale for Localization Branch.** In object detection, to balance the training of the localization and classification branches, traditional detectors [31] usually add a loss weight to the localization branch. While the Parameterized AP Loss is a unified loss, the network itself still preserves independent branches for these two tasks. In order to balance the training weights, we multiply the gradient back-propagated through the localization branch with a gradient scale $\lambda > 0$ (see Figure 1), which serves similar effect with the traditional loss weight.

In practice, the actual searched parameter is $\theta_\lambda = \frac{1}{2}(\log_{10} \lambda + 1)$. Such search parameterization form simplifies the search for small and large gradient scale $\lambda$ values. We also empirically restrict the search range for $\theta_\lambda$ as $(0, 1)$, *i.e.*, $\lambda \in (0.1, 10)$.

The collection of the searched parameter for the gradient scale $\theta_\lambda$ and other parameters in the Parameterized AP Loss $\{\theta_i\}_{i=1}^5$ are denoted as $\Theta$, which could be optimized by efficient parameter search methods.

### 3.3 Searching for Optimal Parameters

The optimal parameter set $\Theta$ for the Parameterized AP Loss is found by optimizing the network performance on the validation set with the AP metric. The massive parameter space of $\Theta$ makes it impractical to determine the desired parameters manually. Efficient automatic search is thus necessary to find the optimal parameters.

Our search process is described in Algorithm 1. Specifically, we divide the training set into two subsets in the search process, $\mathcal{D}_{\text{train}}$ for training and $\mathcal{D}_{\text{eval}}$ for evaluation, respectively. The whole search process is treated as a bi-level optimization problem, which can be formulated as

$$\max_\Theta \quad R(\Theta) = \text{AP}(\omega^*(\Theta); \mathcal{D}_{\text{eval}})$$
$$\text{s.t.} \quad \omega^*(\Theta) = \min_\omega L_\Theta(\omega; \mathcal{D}_{\text{train}}), \quad (9)$$

where $L_\Theta$ is the Parameterized AP Loss with the loss parameter set $\Theta$, $\omega$ stands for the network weights, and $\omega^*(\Theta)$ indicates the trained network weights with the given loss parameterized with $\Theta$. $\text{AP}(\omega; \mathcal{D})$ evaluates the AP metric for the network with weights $\omega$ on the given dataset $\mathcal{D}$.

To optimize Eq. (9), we optimize the inner level and outer level problem iteratively. At the inner level, we train a detector network on $\mathcal{D}_{\text{train}}$ with the certain loss parameter set $\Theta$ for one epoch. At the outer level, following [43, 18], we adopt reinforcement learning, a commonly used hyper-parameter optimization algorithm, to search for the optimal parameters. Other efficient search methods like evolutionary algorithm may also be applied. Specifically, we adopt the PPO2 [39] algorithm. Here, we consider a search process consisting of $T$ rounds. In the $t$-th round, we sample $S$ parameter set

---

**Algorithm 1:** Parameterized AP Loss Search Process

---

**Input:** Network initial weights $\omega_0$, initial parameter set distribution $(\mu_1, \sigma^2 I)$, training dataset $\mathcal{D}_{\text{train}}$ and evaluation dataset $\mathcal{D}_{\text{eval}}$ for the proxy task

**Output:** Searched optimal parameter set $\Theta^*$

**for** $t \leftarrow 1$ **to** $T$ **do**

    **for** $i \leftarrow 1$ **to** $S$ **do**

        Sample parameter set $\Theta_i^t \sim N_{trunc[0,1]}(\mu_t, \sigma^2 I)$;

        Inner-level network training with the initial network weights $\omega_0$,

        $\omega^*(\Theta_i^t) = \min L_\Theta(\omega; \mathcal{D}_{\text{train}})$;

        Evaluate the AP metric for the trained network as the corresponding reward,

        $R(\Theta_i^t) = \text{AP}(\omega^*(\Theta_i^t); \mathcal{D}_{\text{eval}})$;

    **end**

    Outer-level distribution update, $\mu_{t+1} = \underset{\mu}{\text{argmax}} \frac{1}{S} \sum_{i=1}^{S} \tilde{R}_i^t(\mu, \mu_t, \Theta_i)$;

**end**

**return** $\Theta^* = \text{argmax}_\Theta R(\Theta_i^t), \quad \forall t = 1, \ldots, T, \ i = 1, \ldots, S$

---

samples $\{\Theta_i^t\}_{i=1}^{S}$ independently from a truncated normal distribution [4] $N_{trunc[0,1]}(\mu_t, \sigma^2 I)$, where $\mu_t$ and $\sigma^2$ are the mean and variance values, respectively. The truncated range of the distribution is set to $[0, 1]$ so as to satisfy the monotonicity constraint of independent parameters in Eq. (8). For the $i$-th sample, the AP metric score of the network trained in the inner level is regarded as its reward $R_i^t = R(\Theta_i^t)$. In the PPO2 algorithm, the mean value of the truncated normal distribution for the next $(t + 1)$-th round is updated as

$$\mu_{t+1} = \underset{\mu}{\text{argmax}} \frac{1}{S} \sum_{i=1}^{S} \tilde{R}_i^t(\mu, \mu_t, \Theta_i^t). \tag{10}$$

Here $\tilde{R}_i^t(\mu, \mu_t, \Theta_i)$ is calculated from the original reward $R_i^t$ of each sample as

$$\tilde{R}_i^t(\mu, \mu_t, \Theta_i) = \min \left( \frac{p(\Theta_i^t; \mu, \sigma^2 I)}{p(\Theta_i^t; \mu_t, \sigma^2 I)} R_i^t, \text{CLIP} \left( \frac{p(\Theta_i^t; \mu, \sigma^2 I)}{p(\Theta_i^t; \mu_t, \sigma^2 I)}; 1 - \epsilon, 1 + \epsilon \right) R_i^t \right), \tag{11}$$

where $\min(\cdot, \cdot)$ outputs the smaller value between two inputs, $p(\Theta_i^t; \mu, \sigma^2 I)$ denotes the PDF of given truncated normal distribution, and the CLIP function clips input value within $1 - \epsilon$ and $1 + \epsilon$ as stated in PPO2 [39] for more stable and effective search. Note that the mean reward in the $t$-th round is subtracted from each $R_i^t$ for better convergence. More implementation details are described in Section 4.1.

## 4 Experiments

We evaluate our approach on the COCO 2017 object detection benchmark[2] [23] with various object detectors, including RetinaNet [25], Faster R-CNN [37] and Deformable DETR [42]. Note that for Faster R-CNN, we search the losses for both the RPN and R-CNN heads simultaneously.

### 4.1 Implementation Details

**Loss Function Calculation.** As described in Section 3.2, the inputs to the parameterized functions $f(x; \theta)$ are normalized to range $[0, 1]$ for a unified domain. For classification score differences $s_j - s_i$, because their original values are unbounded, we first clip them within the range of $[-1, 1]$ following [5, 30]. The clipped values are further mapped to the range of $[0, 1]$ via min-max re-scaling. For the localization scores $l(b_i)$, GIoU [38] is used as the measurement by default due to its good performance. The localization scores are also mapped to the range of $[0, 1]$ via min-max re-scaling. The default value of number of segments $M$ is fixed as 5. In experiments, the whole prediction set $\mathcal{B}$ in Eq. (6) consists of all predicted boxes in the current mini-batch for all categories. The positive

---

[2]COCO 2017 is publicly available under the Creative Commons Attribution 4.0 License. As far as we know, it does not contain any personally identifiable information or offensive content.

**Table 1:** Performance of different losses on the COCO benchmark.

| Model | Loss | AP | $AP_{50}$ | $AP_{75}$ | $AP_S$ | $AP_M$ | $AP_L$ |
|---|---|---|---|---|---|---|---|
| RetinaNet [25]
ResNet-50 [12] + FPN [24] | Focal Loss [25] + L1 | 37.5 | 57.3 | 39.5 | 20.4 | 41.9 | 52.2 |
| | Focal Loss [25] + GIoU [38] | 39.2 | 58.1 | 41.4 | 22.0 | 43.6 | 53.3 |
| | AP Loss [5] + L1 | 35.4 | 58.1 | 37.0 | 19.2 | 39.7 | 49.2 |
| | aLRP Loss [30] | 39.0 | 58.7 | 40.7 | 22.0 | 43.5 | 54.0 |
| | Parameterized AP Loss (ours) | **40.5** | **59.0** | **43.4** | **23.9** | **44.9** | **56.1** |
| Faster R-CNN [37]
ResNet-50 [12] + FPN [24] | Cross Entropy + L1 | 39.0 | 59.6 | 42.2 | 22.4 | 43.5 | 53.2 |
| | Cross Entropy + GIoU [38] | 39.1 | 59.3 | 42.4 | 23.2 | 43.4 | 53.5 |
| | aLRP Loss [30] | 40.7 | 60.7 | 43.3 | 23.2 | 45.2 | 56.6 |
| | AutoLoss-Zero [19] | 39.3 | 59.0 | 42.4 | 21.4 | 44.0 | 54.0 |
| | CSE-AutoLoss-A [28] | 40.4 | 60.6 | 43.8 | 23.8 | 45.0 | 55.7 |
| | Parameterized AP Loss (ours) | **42.0** | **60.7** | **45.0** | **25.3** | **46.6** | **57.7** |
| Deformable DETR [42]
ResNet-50 [12] | Focal Loss [25] + L1 + GIoU [38] | 43.8 | 62.6 | 47.7 | 26.4 | 47.1 | 58.0 |
| | Parameterized AP Loss (ours) | **45.3** | **63.1** | **49.6** | **27.9** | **49.3** | **60.2** |

prediction set $\mathcal{P}$ and the negative prediction set $\mathcal{N}$ are determined by the original pre-defined training target assignment of each object detector.

**Search Settings.** In COCO 2017 [23], there are 118k images in the train subset. As described in Section 3.3, we randomly divide the original train subset into $\mathcal{D}_{\text{train}}$ and $\mathcal{D}_{\text{eval}}$, which constitute 113k training images and 5k evaluation images for the proxy task, respectively. In the inner-level network training of Algorithm 1, we train the object detectors for one epoch. For Deformable DETR [42], we also set the number of feature levels to 1 to further accelerate the training. In the outer-level PPO2 [39] updating of Algorithm 1, we sample $S = 8$ samples each round, and search for $T = 40$ rounds in total. The mean vector $\mu_1$ of the truncated normal distribution is initialized to make $f(x; \theta) = x$. The standard deviation $\sigma$ is initialized as $0.2$, which decays linearly to 0 with respect to the search round. The clip operation in Eq. (11) is applied on each component value in $\Theta$ independently. The clip range $\epsilon$ is set to $0.1$ following PPO2 [39]. To solve Eq. (10), we employ the Adam optimizer [16] for 100 iterations with a base learning rate of $0.01$. In the first 30 iterations, linear warm-up from a learning rate of 0 is utilized.

**Training Settings.** After the parameter search, we re-train the object detectors with the searched Parameterized AP Loss on the COCO 2017 train subset, and evaluate them on the val subset, which consists of 5k images. For RetinaNet [25] and Faster R-CNN [37], ImageNet [7] pre-trained ResNet-50 [12] with FPN [24] is utilized as the backbone, and the training settings strictly follow [5, 30]. For Faster R-CNN, we use a base learning rate of $0.024$ and a batch size of $64$ (8 images on each GPU). For RetinaNet, the base learning rate is set to $0.016$ and the batch size is $64$ (8 images on each GPU). For Deformable DETR, ImageNet [7] pre-trained ResNet-50 [12] is utilized as the backbone, and the training settings strictly follow [42], where the learning rate is set to $0.0002$ and the batch size is $32$ (4 images on each GPU). All the experiments are conducted on 8 NVIDIA V100 GPUs. Our method is implemented based on the open-sourced MMDetection codebase[3] [6].

### 4.2 Main Results

Table 1 summarizes the performance of our approach and the existing losses on COCO 2017 [23]. Among these existing losses, Cross Entropy and Focal Loss [25] are commonly used for classification, while L1 and GIoU [38] are commonly used for localization. AP Loss [5] is a classification loss handcrafted for error-driven optimization of the AP metric, while aLRP Loss [30] is an error-driven method specifically designed for the LRP metric [29]. Compared with these handcrafted losses, Parameterized AP Loss can yield $1.5 \sim 3.0$ AP score gain. We also compared with AutoLoss-Zero [19] and CSE AutoLoss [28], which are two AutoML-based losses. Our approach can obtain over $1.5$ AP score improvement over them. We also seek to compare with the direct optimization methods [13, 40, 35], but their codes are not released and their original training settings differ too much from ours, which makes them difficult to be compared with. The searched parameterized functions $f(x; \theta)$ of our approach are demonstrated in Appendix A.1.

---

[3]MMDetection is an open-sourced codebase under the Apache-2.0 License.

### 4.3 Ablation Study

In this section, we ablate different design choices and the search effectiveness. RetinaNet [25] is employed as the baseline model in the following experiments.

**Comparing Searched and Hand-crafted Substitutions for $H(\cdot)$.** Here, we compare our searched parameterized function $f(\cdot; \theta)$ with several handcrafted differentiable substitutions, including sigmoid, sqrt, linear, and square functions. Table 2 shows that the searched parameterized function demonstrates significant improvement. The low performance of hand-crafted substitutions indicates that it is non-trivial to hand-craft appropriate differentiable approximations. As analyzed in [5], the sigmoid substitution for $H(\cdot)$ actually may lead to non-convergence of the network training. Note that the linear function is also the initialization of our searched parameterized function, which indicates the effectiveness of the search process.

**Comparing Separate and Shared Parameters for Different $H(\cdot)$.** As stated in Section 3.2, we use separate parameters for different $H(\cdot)$ functions. We have also tried using shared parameters for all the $H(\cdot)$ functions. Table 3 shows that using separate parameters can yield nearly 3.0 AP improvement. That is because separate parameters can enhance the flexibility, so that different components in Eq. (6) can search for its own shape to better adjust the loss function.

**Comparing with and without Gradient Scale $\lambda$.** As described in Section 3.2, the searched gradient scale is adopted to serve similar effect as the traditional loss weight. We have also tried not using the gradient scale, *i.e.,* $\lambda$ is fixed as 1. Table 4 shows that the automatically searched gradient scale can indeed improve the performance.

**Comparing Different Measurements for Localization Score $l(b_i)$.** Table 5 shows the results of using L1, IoU and GIoU [38] for the localization score. GIoU and L1 produce similar results, both of which outperform the performance of IoU. We argue that this is because IoU will back-propagate meaningful gradients only if two boxes are overlapped, while GIoU and L1 do not have such problem.

**Comparing Different Number of Segments in $f(\cdot; \theta)$.** Table 6 shows the results. Too less segments will limit the expressiveness of $f(\cdot; \theta)$, which leads to significant drop in performance. However, too many segments may increase the search difficulty. In practice, 5 segments is enough.

**Comparing PPO2 [39] and Random Search.** Figure 3 shows that PPO2 can find better parameters more efficiently than random search, which suggests that loss function search is non-trivial and reinforcement learning helps to accelerate the search process.

**Table 2:** Comparison of the searched parameterized function and the hand-crafted substitutions for $H(\cdot)$.

| Differentiable Substitution | AP | AP$_{50}$ | AP$_{75}$ | AP$_S$ | AP$_M$ | AP$_L$ |
|---|---|---|---|---|---|---|
| Sigmoid | 3.2 | 5.9 | 2.8 | 2.7 | 4.0 | 4.9 |
| Sqrt | 2.3 | 4.3 | 2.2 | 1.8 | 2.9 | 4.0 |
| Linear | 22.9 | 39.3 | 23.2 | 16.9 | 27.3 | 29.2 |
| Square | 36.4 | 56.5 | 39.9 | 20.3 | 41.7 | 53.0 |
| Searched | 40.5 | 59.0 | 43.4 | 23.9 | 44.9 | 56.1 |

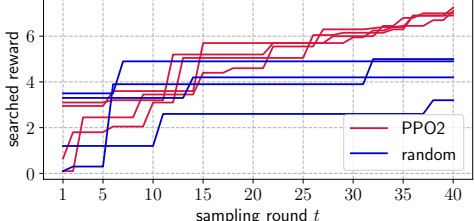

**Figure 3:** Comparison of PPO2 and random search. The search is repeated four times. Each curve presents the highest searched reward up to the $t$-th round.

**Table 3:** Comparison of shared and separate parameters for different $H(\cdot)$.

| Parameters | AP | AP$_{50}$ | AP$_{75}$ | AP$_S$ | AP$_M$ | AP$_L$ |
|---|---|---|---|---|---|---|
| Shared | 37.8 | 58.3 | 39.9 | 21.6 | 42.2 | 52.1 |
| **Separate** | 40.5 | 59.0 | 43.4 | 23.9 | 44.9 | 56.1 |

**Table 4:** Comparison of our proposed loss with and without the gradient scale $\lambda$.

| Gradient Scale | AP | AP$_{50}$ | AP$_{75}$ | AP$_S$ | AP$_M$ | AP$_L$ |
|---|---|---|---|---|---|---|
| w/o | 39.4 | 58.9 | 42.0 | 22.8 | 43.9 | 54.2 |
| **w/** | 40.5 | 59.0 | 43.4 | 23.9 | 44.9 | 56.1 |

**Table 5:** Comparison of different measurements for the localization score $l(b_i)$.

| Measurement | AP | AP$_{50}$ | AP$_{75}$ | AP$_S$ | AP$_M$ | AP$_L$ |
|---|---|---|---|---|---|---|
| L1 | 40.0 | 58.8 | 43.2 | 22.3 | 44.5 | 55.3 |
| IoU | 38.1 | 56.2 | 40.8 | 21.8 | 43.6 | 53.4 |
| **GIoU** | 40.5 | 59.0 | 43.4 | 23.9 | 44.9 | 56.1 |

**Table 6:** Comparison of different number of segments in the piecewise linear functions $f(\cdot; \theta)$.

| Segments | AP | AP$_{50}$ | AP$_{75}$ | AP$_S$ | AP$_M$ | AP$_L$ |
|---|---|---|---|---|---|---|
| 3 | 34.2 | 48.3 | 36.8 | 9.2 | 43.8 | 56.9 |
| **5** | 40.5 | 59.0 | 43.4 | 23.9 | 44.9 | 56.1 |
| 7 | 40.3 | 58.7 | 43.2 | 23.8 | 44.6 | 55.9 |

# 5    Conclusion

In this paper, we proposed Parameterized AP Loss, which represents numerous AP approximations in a unified formula, and automatically searches for the optimal loss function. Parameterized AP Loss is a single unified loss function capturing the classification and localization sub-tasks simultaneously, which consistently outperforms existing loss functions on various object detectors. Although we have verified the effectiveness of parameterization in searching for optimal loss functions for the AP metric, there are still open questions about whether such technique can be extended to other non-differentiable metrics in different tasks, such as the widely used BLEU metric in machine translation, where the calculation of n-gram might be non-trivial to be parameterized.

**Potential Negative Societal Impacts.**  Our searched losses share the same societal issues with other hand-crafted ones, that the trained object detectors may have inexplicable detection failures and suffer from data bias. The automatic search process in our method is laborsaving, which may also have a negative impact on social employment opportunities.

## Acknowledgments and Disclosure of Funding

The work is supported by the National Key R&D Program of China (2020AAA0105200) and Beijing Academy of Artificial Intelligence.

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
