# A  Appendix

## A.1  More Ablations and Visualizations

**Effect of Blocking Gradient of $f(s_j - s_i; \theta_4)$.** As mentioned in Section 3.2, we compare the performance of different detectors with or without blocking the gradient of $f(s_j - s_i; \theta_4)$ on the COCO benchmark [5] in Table 1. These results indicate that blocking the gradient of $f(s_j - s_i; \theta_4)$ can greatly boost the performance. We attribute this to the unstable training caused by the gradient from the denominator, so they are blocked out by default in the experiments.

**Visualization of Searched Parameterized Functions.** Figure 1 visualizes the searched parameterized functions for different detectors on the COCO benchmark [5]. Each line corresponds to an independent parameterized function in Eq. (6). The dots on each line represent the control points for each parameterized function. It can be observed that loss functions for different detectors seem to differ from each other. Their intrinsic differences can lead to distinct loss functions.

**Parameterized AP Loss on PASCAL VOC Benchmark [3].** We also search Parameterized AP Loss on the PASCAL VOC benchmark and compare its performance with the commonly used Focal Loss [7] and L1 combination. We adopt RetinaNet [7] with ImageNet [2] pre-trained ResNet50 [4] and FPN [6] as the backbone. The training setting strictly follows the default config in MMDetection codebase [1]. During search, we train the object detector for one epoch as the proxy task. Table 2 shows that the searched loss can perform well on the PASCAL VOC benchmark, bringing around 3.0 $AP_{50}$ improvement. We also search Parameterized AP Loss on the COCO benchmark, and use the searched loss to guide the training on PASCAL VOC benchmark. The result is superior than the Focal Loss and L1 combination, which shows that the searched loss has a certain generalization ability among different datasets.

**More Detailed Ablations on Differentiable Substitution and Gradient Scale.** Table 3 gives a more detailed study on how substitution and gradient scale affect performance. It's shown that even without searched gradient scale and separate parameterization, Parameterized AP Loss  can still outperform handcrafted substitutions. Separate parameterization can bring more than 2.0 AP gain, and gradient scale can further boost the performance by around 1.0 AP.

**Table 1:** The effect of blocking gradient of $f(s_j - s_i; \theta_4)$ on the COCO benchmark.

| Model | Block Gradient | AP | $AP_{50}$ | $AP_{75}$ | $AP_S$ | $AP_M$ | $AP_L$ |
|---|---|---|---|---|---|---|---|
| RetinaNet [7] | | 0.8 | 1.3 | 0.9 | 0.5 | 0.9 | 1.2 |
| ResNet-50 [4] + FPN [6] | ✓ | 40.5 | 59.0 | 43.4 | 23.9 | 44.9 | 56.1 |
| Faster R-CNN [8] | | 29.5 | 42.4 | 31.4 | 14.4 | 32.3 | 42.6 |
| ResNet-50 [4] + FPN [6] | ✓ | 42.0 | 60.7 | 45.0 | 25.3 | 46.6 | 57.7 |
| Deformable DETR [9] | | 27.8 | 54.9 | 25.8 | 16.8 | 31.5 | 35.6 |
| ResNet-50 [4] | ✓ | 45.3 | 63.1 | 49.6 | 27.9 | 49.3 | 60.2 |

**Table 2:** Comparison on the PASCAL VOC benchmark with RetinaNet.

| Loss | $AP_{50}$ |
|---|---|
| Focal Loss [7] + L1 | 77.3 |
| Parameterized AP Loss searched on COCO | 79.6 |
| Parameterized AP Loss searched on PASCAL VOC | 80.2 |

**Table 3:** More detailed ablations on differentiable substitution and gradient scale.

| Differentiable Substitution | Gradient Scale | AP | $AP_{50}$ | $AP_{75}$ | $AP_S$ | $AP_M$ | $AP_L$ |
|---|---|---|---|---|---|---|---|
| Sigmoid | 1 | 3.2 | 5.9 | 2.8 | 2.7 | 4.0 | 4.9 |
| Sqrt | 1 | 2.3 | 4.3 | 2.2 | 1.8 | 2.9 | 4.0 |
| Linear | 1 | 22.9 | 39.3 | 23.2 | 16.9 | 27.3 | 29.2 |
| Square | 1 | 36.4 | 56.5 | 39.9 | 20.3 | 41.7 | 53.0 |
| Searched Shared | 1 | 37.1 | 58.1 | 39.1 | 21.0 | 41.7 | 51.1 |
| Searched Separate | 1 | 39.4 | 58.9 | 42.0 | 22.8 | 43.9 | 54.2 |
| Searched Shared | searched | 37.8 | 58.3 | 39.9 | 21.6 | 42.2 | 52.1 |
| Searched Separate | searched | 40.5 | 59.0 | 43.4 | 23.9 | 44.9 | 56.1 |

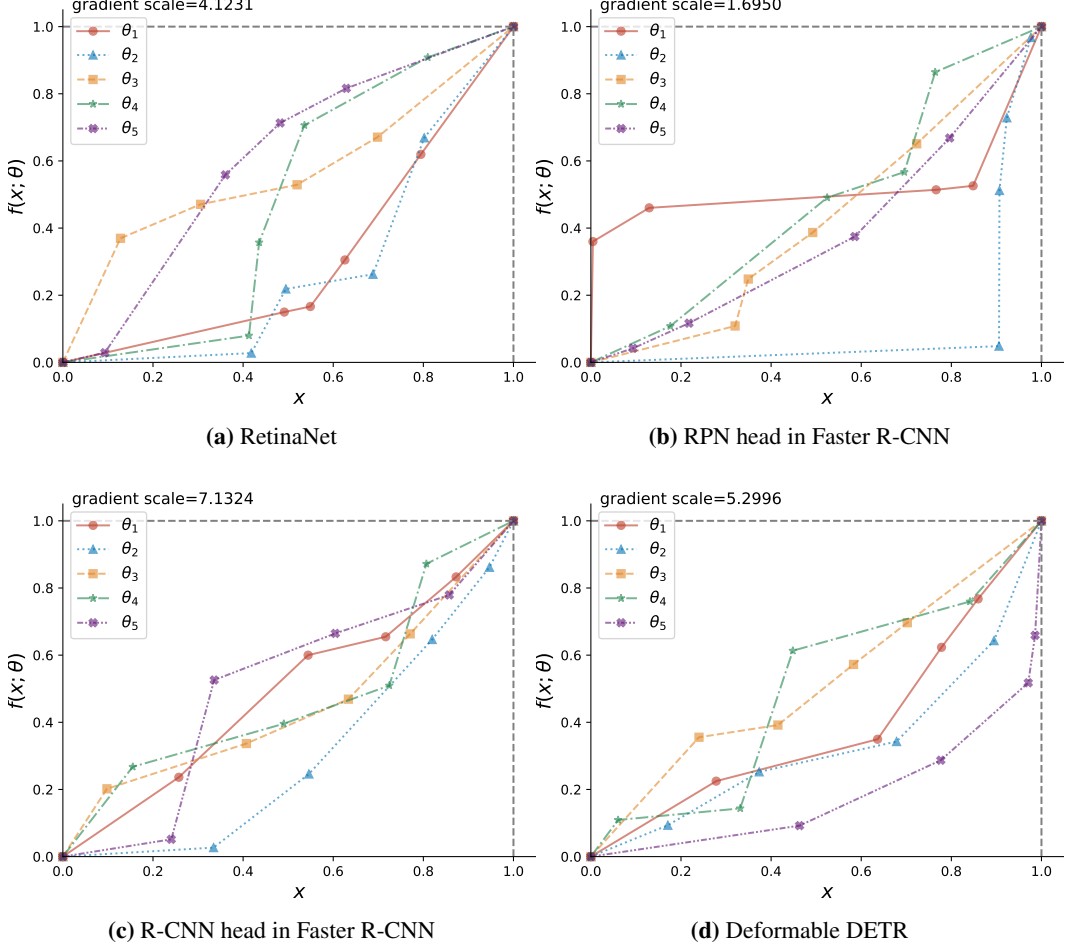

**(a)** RetinaNet

**(b)** RPN head in Faster R-CNN

**(c)** R-CNN head in Faster R-CNN

**(d)** Deformable DETR

**Figure 1:** Visualization of the searched parameterized functions on the COCO benchmark.