# OpenReview forum: "Searching Parameterized AP Loss for Object Detection"
_NeurIPS.cc/2021/Conference — NeurIPS 2021 Poster_

### Official Review · Reviewer_L4vJ · 2021-07-11

**Rating:** 7
**Confidence:** 3

**Summary:**

This paper proposes a parameterized AP loss. Parameterizd functions are introduced to substitute the non-differentiable components in the AP calculation. Then automatic parameter search algorithms can be employed to search the optimal parameters.

**Limitations And Societal Impact:**

The searched losses in this paper share the same societal issues with other hand-crafted ones, that the trained object detectors may have inexplicable detection failures and suffer from data bias. This problem mainly needs to be solved from the perspective of dataset.

**Main Review:**

The method proposed by this paper is novel to some extent and technically sound. The written is also clear.

Strong points:

The parameterized AP Loss can produce better performance, while hand-crafted AP approximated loss produce lower performance than the commonly used cross-entropy and L1/smooth-L1 losses.

Weak points:

1.Parameterization makes the optimal parameters of the proposed AP Loss different on different datasets, so it is necessary to search for the optimal parameters for each dataset. However, the process of finding optimal parameters is computationally expensive.

2.The AP loss proposed in this paper uses a series of parameterized functions to approximate the calculation of AP, but this paper does not show whether the finally searched functions are really close to the Heaviside step function. If the searched function curve can be drawn, it will give readers a more intuitive understanding.


**Time Spent Reviewing:**

3

---

> ### Author Response · Authors · 2021-08-10
> **Reply to Reviewer L4vJ**
>
> Thanks for the thorough and constructive comments. We clarify the questions as follows.
>
> -----
> Q#1: Parameterization makes the optimal parameters of the proposed AP Loss different on different datasets, so it is necessary to search for the optimal parameters for each dataset. However, the process of finding optimal parameters is computationally expensive.
>
> A#1: The searched optimal parameters can generalize to different datasets. Thus, it's not necessary to search again for each new dataset. We have evaluated the performance on the PASCAL VOC benchmark with loss searched on COCO, and the result outperforms the Focal Loss + L1 Loss combination, which proves the generalization ability. We shall add this experiment to the revised version.
>
> |Loss | $\mathrm{AP}_\mathrm{50}$ on PASCAL VOC |
> |:----|:----:|
> |Focal Loss + L1| 77.3 |
> |Parmaeterized AP Loss searched on COCO | 79.6 |
>
> -----
> Q#2: If the searched function curve can be drawn, it will give readers a more intuitive understanding.
>
> A#2: We have visualized the searched parameters for each object detector in the supplementary material (see L8-12 in supplementary).

---

> > ### Comment · Reviewer_L4vJ · 2021-08-15
> > **Reply: Reply**
> >
> > Thank you for your reply. Since the searched parameters have a certain generalization ability, I think overall this is a good paper.

---

### Official Review · Reviewer_sg2b · 2021-07-15

**Rating:** 6
**Confidence:** 5

**Summary:**

This paper proposes a Parameterized AP Loss to better align the network training and evaluation in object detection. The main technical contributions include:
1) It proposes a parameterized loss function by reformulating the AP metric and replacing the non-differentiable component with parameterized functions.
2) It automatically searches for the optimal parameters of the proposed loss function through a reinforcement-learning-based search process.
3) Finally, it demonstrates competitive results on COCO.


**Limitations And Societal Impact:**

Yes.

**Main Review:**

Pros:
1. The idea of introducing localization score into AP loss and parameterizing the reformulated AP loss is novel and interesting.
2. The paper is clearly written and easy to follow with detailed equations.
3. The authors conducted adequate experiments and ablation studies, verifying the efficacy of their method.
Cons:
There is no big flaw in this paper, here are some small issues:
1. In lines 285-292 and Table 2, what is the value of the gradient scale $\lambda$ used in the experiments with hand-crafted substitutions? The experiments with hand-crafted substitutions share H(.), what is the performance of our method when using the same $lambda$ and shared parameters of H(.) as the hand-crafted ones?
2. What is the difference of the shape of the five parameterized functions $f(x; \theta_{i})$ searched by our method. Will they change dramatically in different object detectors (e.g., RetinaNet, Faster R-CNN, and Deformable DETR)? It would be better to visualize the shape of the parameterized functions.
3. How is the generalization of the Parameterized AP Loss. Does it also achieve a good performance when using the hyper-parameters searched on different datasets? For example, searching the hyper-parameters on COCO and using the same hyper-parameters to train the object detector on VOC.


**Time Spent Reviewing:**

1.5

---

> ### Author Response · Authors · 2021-08-10
> **Reply to Reviewer sg2b**
>
> Thanks for the thorough and constructive comments. We clarify the questions as follows.
>
> -----
> Q#1: In lines 285-292 and Table 2, what is the value of the gradient scale $\lambda$ used in the hand-crafted substitutions? What is the performance of our method when using the same $\lambda$ and shared parameters of $H(\cdot)$ as the hand-crafted ones?
>
> A#1: We set the gradient scale $\lambda$ for the hand-crafted substitutions as 1 (i.e., without gradient scale). The performance of our searched functions with/without gradient scale and using shared/separate parameterization are reported below. Even using shared parameters and without gradient scale, the searched parameterization can still outperform hand-crafted substitution. The separate parameterization and searched optimal gradient scale can further boost the performance. We shall add these experiment results in the revised version.
>
> |Differentiable Substitution | gradient scale | $\mathrm{AP}$ | $\mathrm{AP}_\mathrm{50}$ | $\mathrm{AP}_\mathrm{75}$ | $\mathrm{AP}_\mathrm{S}$ | $\mathrm{AP}_\mathrm{M}$ | $\mathrm{AP}_\mathrm{L}$ |
> |:----:|:----:|:----:|:----:|:----:|:----:|:----:|:----:|
> |Sigmoid | 1 | 3.2 | 5.9 | 2.8 | 2.7 | 4.0 | 4.9 |
> |Sqrt | 1 | 2.3 | 4.3 | 2.2 | 1.8 | 2.9 | 4.0 |
> |Linear | 1 | 22.9 | 39.3 | 23.2 | 16.9 | 27.3 | 29.2 |
> |Square | 1 | 36.4 | 56.5 | 39.9 | 20.3 | 41.7 | 53.0 |
> |Searched Shared | 1 | 37.1 | 58.1 | 39.1 | 21.0 | 41.7 | 51.1|
> |Searched Separate | 1 | 39.4 | 58.9 | 42.0 | 22.8 | 43.9 | 54.2 |
> |Searched Shared | searched | 37.8 | 58.3 | 39.9 | 21.6 | 42.2 | 52.1|
> |Searched Separate | searched | 40.5 | 59.0 | 43.4 | 23.9 | 44.9 | 56.1 |
>
> -----
> Q#2: What is the difference of the shape of the five parameterized functions searched by our method. It would be better to visualize the shape of the parameterized functions.
>
> A#2: We have visualized the searched parameters for each object detector in the supplementary material (see L8-12 in supplementary). It can be observed that loss functions for different detectors seem to differ from each other. Their intrinsic differences can lead to distinct loss functions.
>
> -----
> Q#3: How is the generalization of the Parameterized AP Loss. Does it also achieve a good performance when using the hyper-parameters searched on different datasets? For example, searching the hyper-parameters on COCO and using the same hyper-parameters to train the object detector on VOC.
>
> A#3: The searched hyper-parameters can generalize on different datasets. The Parameterized AP loss searched on COCO for RetinaNet performs well on the PASCAL VOC benchmark comparing to Focal Loss + L1 Loss combination. We shall add this experiment to the revised version.
>
> |Loss | $\mathrm{AP}_\mathrm{50}$ on PASCAL VOC |
> |:----|:----:|
> |Focal Loss + L1| 77.3 |
> |Parmaeterized AP Loss searched on COCO | 79.6 |

---

### Official Review · Reviewer_8qFe · 2021-07-15

**Rating:** 6
**Confidence:** 4

**Summary:**

Authors propose a loss for object detection, that is a relaxation of mAP suitable for use in gradient ascent. Where AP includes Heaviside (step) functions, the authors replace these by learnt piecewise-linear functions. These functions are trained to maximise AP on a held-out set achieved by a detection model that was itself trained using them as the loss (so they do not aim to approximate AP closely, but to derive a surrogate loss that is 'structurally similar' to AP and achieves good training performance). Experiments on COCO 2017 show accuracy improvements over baselines (trained with focal/XE + regression losses) and other techniques approximating AP.

**Limitations And Societal Impact:**

There is only very minimal discussion of limitations of the method. Societal impact is discussed sufficiently.

**Main Review:**

Strengths:

The proposed method is reasonably well-motivated, in terms of limitations of existing methods that do not optimise the surrogate itself to be 'good' as a training loss.

The precise structure of the proposed surrogate loss is novel; also I don't know of any existing methods for object detection that aim to tune the surrogate.

Empirical results indicate some improvement over earlier methods; these gains are consistent across various different detection architectures. There is also a detailed ablation study. There is plenty of detail on the experimental details and training protocols.


Concerns / suggestions:

There are no error bars / standard deviations on the main experiments. This makes it difficult to judge how much better the proposed method really is.

By replacing the Heaviside functions with an arbitrary monotonic function, the loss ceases to be AP; rather, it becomes a proxy loss chosen to yield high test-set AP. But given this, there is not really any justification for retaining the full structure of AP in the loss -- larger chunks of eq. 2 than just the H(...) terms could be replaced. What is the justification for not doing so?

"A Unified Framework of Surrogate Loss by Refactoring and Interpolation" [Liu, ECCV 20] is very relevant, and should be cited (and either evaluated against, or discussed why it is not applicable).

"Learning Surrogates via Deep Embedding" [Patel, ECCV 20] is relevant and should be cited.

"SoDeep: a Sorting Deep net to learn ranking loss surrogates" [Engilberge, CVPR 19] is relevant and should be cited.

It would be useful to add some insights on why the various learnt piecewise-linear functions look so different.

What happens if the Heaviside functions are instead replaced by sigmoids of varying slope/offset? This results in many fewer parameters in the loss, but gives more flexibility than just sigmoid.

There is no discussion of how long the PPO training of the surrogate loss takes (e.g. does it dominate the main detector training time)?

The same training process used for the loss parameters (i.e. PPO2) could, theoretically, also be used to train the main detection model. There should be some discussion of why this is not done (or even better, an evaluation).

42: "helps to represent the numerous AP approximations" -- clarify -- is this supposed to refer back to the AP approximations discuss in the previous bit of text?

46: "more effective" -- than what?

---

## Post-rebuttal

The authors' comment clarified several concerns raised in my review (e.g. the search does not need to be repeated for each dataset; the standard deviations are small). I therefore still broadly favor acceptance. However, the rebuttal simply says that UniLoss [Liu, ECCV '20] -- which is one of the most relevant recent works -- will be discussed in the final version; there is no attempt to justify why it was ignored so far, and no promise to include an evaluation against it. Therefore, I cannot raise my score higher. Concerns raised by the other reviewers seem to have been adequately addressed by the authors' comments.


**Time Spent Reviewing:**

2.5

---

> ### Author Response · Authors · 2021-08-10
> **Reply to Reviewer 8qFe**
>
> Thanks for the thorough and constructive comments. We clarify the questions as follows.
>
> -----
> Q#1: There are no error bars / standard deviations on the main experiments.
>
> A#1: The experiment results are reported following the common practice on COCO benchmark, where the final results are reported without error bars or standard deviations. Experimentally, the AP results on COCO benchmark are quite stable (usually with std < 0.3 AP).
>
> -----
> Q#2: Larger chunks in Eq.2 could be replaced, why not doing so?
>
> A#2: We agree that it is very interesting to replace larger chunks in Eq.2 to see whether the surrogate loss can be searched with less prior of the metric formulation. We shall consider trying this experiment in the future. In this paper, we simply replace the $H(\cdot)$ functions as a preliminary attempt.
>
> -----
> Q#3: Should cite [1,2,3].
>
> A#3: Thanks for your kind reminder. We shall cite and discuss them in the revised version.
>
> [1] Liu et. al., A Unified Framework of Surrogate Loss by Refactoring and Interpolation, ECCV 2020;
>
> [2] Patel et. al., Learning Surrogates via Deep Embedding, ECCV 2020;
>
> [3] Engilberge et. al., SoDeep: a Sorting Deep net to learn ranking loss surrogates, CVPR 2019.
>
> -----
> Q#4: It would be useful to add some insights on why the various learnt piecewise-linear functions look so different.
>
> A#4: Currently, we have little understanding of the searched loss functions. We are trying some analysis to understand better. Experimentally, we found that even for the same object detector and dataset, there are multiple optimal hyperparameters achieving similar performance. This phenomenon makes it more difficult to be understood.
>
> -----
> Q#5: What happens if the Heaviside functions are instead replaced by sigmoids of varying slope/offset?
>
> A#5: It's an interesting idea to try sigmoids of varying slope/offset. We shall consider trying this in the future. As we have mentioned in L180-181, it's not necessary to use the piecewise linear function for parameterization. We adopt it just for simplicity.
>
> -----
> Q#6: There is no discussion of how long the PPO training of the surrogate loss takes.
>
> A#6: The search process is roughly 2 times longer than the normal training. However, the search cost is actually negligible. We found that the searched loss is able to generalize among different datasets, so it's not necessary to conduct a search before each training.
>
> As shown below, the Parameterized AP loss searched on COCO for RetinaNet performs better on the PASCAL VOC benchmark comparing to Focal Loss + L1 Loss combination. We shall add this experiment to the revised version.
>
> |Loss| $\mathrm{AP}_\mathrm{50}$ on PASCAL VOC |
> |:----|:----:|
> |Focal Loss + L1| 77.3 |
> |Parmaeterized AP Loss searched on COCO | 79.6 |
>
> -----
> Q#7: Why not use the same training setting as training the main detection model during search?
>
> A#7: The main difference in the training setting is that we only train the model for one epoch during search, and reduce the number of feature level to 1 for Deformable-DETR loss search (see L250-252 in experiment details). These modifications are adopted to speed up the search process. Other settings strictly follow the normal training process.
>
> -----
> Q#8: L42: "helps to represent the numerous AP approximations" -- clarify -- is this supposed to refer back to the AP approximations discuss in the previous bit of text?
>
> A#8: By saying this, we want to emphasize that our method would optimize over the numerous approximations of the non-differentiable discrete AP metric, which is quite different from existing approaches.
>
> -----
> Q#9: L46: "more effective" -- than what?
>
> A#9: We are sorry for the typo. We will revise it as: "the search process would be very effective".

---

### Official Review · Reviewer_KBgc · 2021-07-15

**Rating:** 7
**Confidence:** 4

**Summary:**

This paper describes a technique for estimating a loss for object detection that is well-aligned with the AP loss used at evaluation time. The work builds on a prior formulation of the AP loss as a function on classification scores, but converts the formulation to account for per-bounding box results, and also replaces the Heaviside function with parametrized, searchable functions. Experiments show the gain from the proposed technique using three backbone detectors, and compares to different prior losses.

Post-rebuttal: The rebuttal addressed my concerns.

**Limitations And Societal Impact:**

Discussed

**Main Review:**

1. The work is well-written and sound. All steps are well-explained and justified. The results are strong and extensive, including detailed ablations that verify the contribution of each method component.

2. The choice of baseline losses seems reasonable, and techniques from each family described in Sec. 2 are included in the experiments, except for the "Direct Optimization for the AP Metric" family-- why are results from one of those methods not included?

3. Some statements are subjective, e.g. L91-92. Others are a bit unclear, i.e. L97-98, 104-105. Some text on the other hand is repetitive, e.g. L168-170 have been previously stated multiple times.

4. The authors explain why they use different theta parameters for different parts of Eq. 6, but one choice especially stands out, e.g. the use of theta_5 as opposed to theta_1 for different occurrences of the exact same l(b_i). How important is this experimentally?

5. A few places in the text indicate the approach relies a bit too much on experimental settings, e.g. L179. In general, using a searchable, tunable metric is great, but is there a risk of overfitting?

6. In general, the proposed approach works well, but is there any cost to pay? Does some part of the method's workings get slower or less stable compared to baseline losses?

7. Minor questions:
- Eq. 7, what is the value of M, i.e. how computed?
- L215, which exact equation is used?

**Time Spent Reviewing:**

1.5

---

> ### Author Response · Authors · 2021-08-10
> **Reply to Reviewer KBgc**
>
> Thanks for the thorough and constructive comments. We clarify the questions as follows.
>
> -----
> Q#1: Why are results from the "Direct Optimization for the AP metric" family not included?
>
> A#1: We have discussed the reason in Section 4.2 (see L278-280). Specifically, we have tried to include these methods, but their codes are not released, and their original training settings differ too much from ours, which makes them difficult to be compared with.
>
> -----
> Q#2: Some statements are subjective, e.g., L91-92. Others are a bit unclear, i.e., L97-98, 104-105. Some text on the other hand is repetitive, e.g., L168-170 have been previously stated multiple times.
>
> A#2: Thanks for the suggestions. We shall revise the paper to elaborate on these statements. Details are listed as follows.
>
> L91-92 will be revised as: "While these works take a step towards direct optimization for the AP metric, they all focus on the gradient for the classification task, ignoring the localization task, which may well be sub-optimal for the optimization of AP metric."
>
> L97-98 will be revised as: "In these methods, loss functions are formulated as computational graphs composed of basic mathematical operators. Since the combination of operators is randomly chosen, the search space is very sparse with a large number of unpromising loss functions."
>
> L104-105 will be revised as: "Since the parameterized search space is continuous, the search process would be more effective than the discrete combinatorial optimization in [25,17]."
>
> L168-170 will be revised as: "In contrast to hand-crafting differentiable surrogates for the AP metric, we substitute the non-differentiable components with parameterized functions."
>
> -----
> Q#3: The use of $\theta_5$ stands out as opposed to $\theta_1$ for different occurrences of the exact same $l(b_i)$. How important is this experimentally?
>
> A#3: This is an interesting idea, and we shall consider trying this experiment. We have tried to use the same parameters for all $H(\cdot)$ functions (see Table3), but have not considered using the same parameters for $H(\cdot)$ functions with the same inputs. Nevertheless, please note that whether to use the same parameters for $H(\cdot)$ functions will not change the search or training cost. Thus, we use independent parameters for more flexible loss form and better performance.
>
> -----
> Q#4: Is there a risk of overfitting?
>
> A#4: We carefully avoid overfitting via dataset split (see L248-250 in experiment details). Specifically, we split the original training set into two subsets for training and evaluation during the search process. The searched loss will be evaluated on the original validation set. Thus, the original validation set is invisible during search.
>
> -----
> Q#5: Is there any cost to pay? Does some part of the method's workings get slower or less stable compared to baseline losses?
>
> A#5: During the network training, our searched loss is slightly slower than the commonly used Focal Loss, but is much faster than the previous SOTA aLRP Loss. On the other hand, please note that the training speed is not the focus of this paper, which aims to search for a surrogate loss better aligned with the AP metric.
>
> -----
> Q#6: What is the value of M in Eq.7?
>
> A#6: The default value of M is fixed as 5, as described in L308 in the ablation study. We shall add its default value to implementation details in the revised version.
>
> -----
> Q#7: Which exact equation is used in L215?
>
> A#7: The AP metric is calculated via the standard AP computation program, i.e., COCO evaluation program.

---

### Decision · Program_Chairs · 2021-09-27

**Decision:**

Accept (Poster)

**Comment:**

All the reviewers have supported the acceptance of this paper, so I recommend accepting as well.